# 3D Gaussian Editing With A Single Image

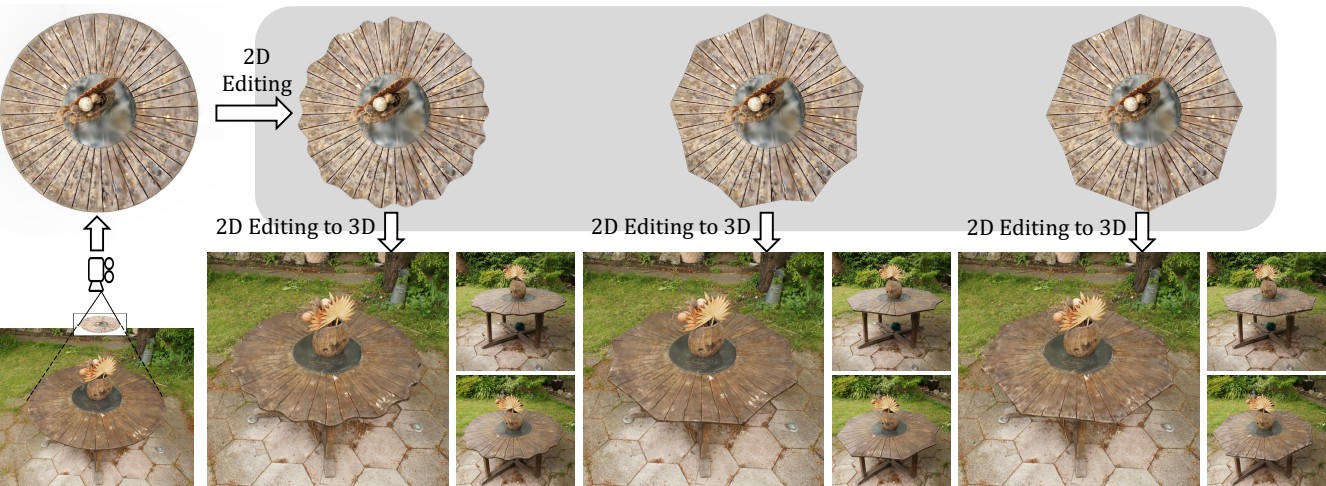

**Figure 1: 3D scene editing with a single image. Given a 3D scene represented by 3D Gaussians and an image edited with 2D editing tools such as PhotoShop, our method can align the underlying scene with the reference image from the specific viewpoint for scene editing, realizing "what you see is what you get", while maintaining overall structural stability.**

## ABSTRACT

The modeling and manipulation of 3D scenes captured from the real world are pivotal in various applications, attracting growing research interest. While previous works on editing have achieved interesting results through manipulating 3D meshes, they often require accurately reconstructed meshes to perform editing, which limits their application in 3D content generation. To address this gap, we introduce a novel single-image-driven 3D scene editing approach based on 3D Gaussian Splatting, enabling intuitive manipulation via directly editing the content on a 2D image plane. Our method learns to optimize the 3D Gaussians to align with an edited version of the image rendered from a user-specified viewpoint of the original scene. To capture long-range object deformation, we introduce positional loss into the optimization process of 3D Gaussian Splatting and enable gradient propagation through reparameterization. To handle occluded 3D Gaussians when rendering from the specified viewpoint, we build an anchor-based structure and employ a coarse-to-fine optimization strategy capable of handling long-range deformation while maintaining structural stability. Furthermore, we design a novel masking strategy that adaptively identifies non-rigid deformation regions for fine-scale modeling.

*ACM MM, 2024, Melbourne, Australia*
© 2024 Copyright held by the owner/author(s). Publication rights licensed to ACM.
ACM ISBN 978-x-xxxx-xxxx-x/YY/MM
https://doi.org/10.1145/nnnnnnn.nnnnnnn

Extensive experiments show the effectiveness of our method in handling geometric details, long-range, and non-rigid deformation, demonstrating superior editing flexibility and quality compared to previous approaches.

## CCS CONCEPTS

• **Computing methodologies → Point-based models**; **Rendering**.

## KEYWORDS

3D Gaussian Splatting, Scene Editing

## 1 INTRODUCTION

3D scene modeling and editing emerge as crucial tools across diverse applications such as film production, gaming, and augmented/virtual reality, offering exceptional advantages. They enable efficient iteration and rapid prototyping, serving as a canvas for creative expression and effective problem-solving. Due to the high laborious cost of traditional mesh-based scene modeling, implicit neural representations, such as neural radiance fields (NeRF), have recently received increasing attention for their lower cost. Although considerable efforts have been made to address the challenge of establishing interpretable connections between visual effects and implicit representations [8, 37, 51, 53, 55], NeRF-based methods still face practical limitations in various applications due to their implicit representation's inability to facilitate explicit manipulation. To significantly enhance the efficiency and quality of 3D scene editing, we represent and edit 3D scenes using the emerging 3D Gaussian Splatting (3DGS) method [22], given its explicit representation and promising reconstruction quality.

Prior neural scene editing methods focus on directly manipulating geometry [51, 53, 55] with the assistance of 3D software, such as Blender. These methods follow a pipeline that extracts meshes from the learned radiance fields and utilizes the geometric structure to guide the deformation of the 3D scene. Due to the imperfect reconstructed geometry, these methods struggle to handle non-rigid deformation and fine-grained editing. Other attempts leverage text-to-image models [2, 18] to edit both the geometry and the texture with text prompts, which are extended to support the manipulation of 3DGS scenes [10, 13]. However, they have a clear limitation: users cannot control the details of the objects in the scene. Unlike previous efforts, our approach is inspired by the way humans observe and perceive the 3D world through 2D images. We introduce a single-image-driven approach to editing the 3D scene, aligning with the philosophy of "what you see is what you get."

In a single-image-driven editing task, the user needs to provide an edited image based on a rendering from a specified viewpoint for the 3D scene. In our work, the 3D scene is reconstructed using 3D Gaussian Splatting [22], and is therefore represented by a set of 3D Gaussian functions. The edited image serves as the target to guide the alignment and manipulation of the 3D content. This process may imply long-range and non-rigid deformation and texture change of 3D objects. We formulate the editing problem as a gradient-based optimization process utilizing 3D Gaussian representation. One trivial solution is to employ photometric losses used in 3DGS [22] to adjust the 3D Gaussians to minimize the difference between the rendered image and the target image. However, these loss functions can only produce intrinsically local derivatives, making them inadequate for handling long-range deformations. Drawing inspiration from DROT [50], we introduce optimal transport into 3D Gaussian optimization to model long-range correspondence explicitly. We propose a positional loss to drive long-range motions and make the overall process differentiable by reparameterization. To ensure the geometric consistency of the objects after editing, we adopt a novel as-rigid-as-possible (ARAP) regularization scheme that operates on a few anchor points to capture the 3D deformation field in a more efficient way. We also design a coarse-to-fine optimization strategy to enhance the fidelity of the edited results. Furthermore, motivated by the observation that objects in the same scene may have different levels of rigidity, we introduce a novel masking strategy to adaptively identify non-rigid deformation parts and release ARAP regularization, enabling more precise modeling of geometric details for real-world scene editing. The contributions of this paper are summarized as follows:

- We propose the first single-image-driven 3D Gaussian scene editing method, realizing "what you see is what you get".
- We introduce positional derivatives into 3DGS to capture long-range deformation and enable gradient propagation through reparameterization.
- We propose an anchor-based as-rigid-as-possible regularization method and a coarse-to-fine optimization strategy to maintain object-level geometry consistency.
- We introduce an adaptive masking strategy to identify non-rigid deformation parts during optimization to ensure more precise modeling.

## 2 RELATED WORK

### 2.1 Differentiable Rendering

Differentiable rendering aims to develop differentiable rendering methods, allowing the computation of derivatives with respect to scene parameters for 3D reconstruction. However, the discontinuities around the object silhouettes pose a significant challenge. To address this issue, [27] introduces an edge sampling method handling Dirac delta functions. SoftRas [29] blurs triangle edges with a signed distance field, aiding gradient back-propagation. [1, 31] approximates boundary terms via reparameterized integrals. The most relevant work to our method is DROT [50], which integrates Optimal Transport into differentiable rendering, explicitly modeling 3D motions through pixel-level correspondence in screen space. Leveraging the correspondence, DROT extends RGB losses with positional loss, ensuring robust convergence in global and long-range object motions.

### 2.2 NeRF and 3D Gaussian Editing

NeRF [34] and its variants [3–7, 15, 35, 47], and 3DGS [22] have gained increasing attention due to their superior view synthesis quality. There is a growing demand for human-friendly editing tools to interact with this representation. [28, 53, 55] proposes to extract meshes from a pre-trained NeRF and edit the 3D scene by manipulating the mesh vertices. [20, 26, 37, 51] simplify the geometry structure by cages and employ a cage-based deformation pipeline for 3D editing. [53] proposes to encode the neural implicit field with disentangled geometry and texture codes on mesh vertices. However, these methods are limited by the quality of the reconstructed geometry and struggle to model non-rigid deformation. [8] mitigates this issue by manipulating feature points, but it is laborious to deal with a large number of feature points. On the other hand, [16, 24, 25] decouple color bases and modify them to achieve texture change, while failing to provide fine-grained editing guidance. [48] adopts a teacher-student knowledge distillation scheme to achieve multi-view appearance consistency. It only supports rigid transformations like rotation and scaling. With the advancement of text-to-image models [38–41], some works [2, 9, 12, 17–19, 33, 42, 43, 46] propose to edit both the geometry and the texture by incorporating CLIP or Diffusion Models to fine-tune NeRF with text instructions. [56] leverages attention maps to locate editing regions. Subsequently, [10, 13, 36, 49] extend semantic editing on NeRFs to 3D Gaussians. However, these methods cannot perform detailed geometry and texture editing. Other works on 3D Gaussian editing [30, 54, 57] involve binding Gaussians to the mesh surface and using the mesh to drive the 3D Gaussians, which are still limited by the quality of the reconstructed meshes. [52] proposes to disentangle geometry and texture for highly efficient texture editing.

## 3 PRELIMINARIES

3D Gaussian Splatting (3DGS) [22] is a recent innovation in neural scene representation, which achieves real-time rendering via splatting 3D Gaussians instead of volumetric rendering. Specifically, it represents the scene as a set of 3D anisotropic Gaussians $\{G_i\}_{i=1}^{N}$, each of which is defined by its center position $\mu_i \in \mathbb{R}^3$, 3D covariance matrix $\Sigma_i \in \mathbb{R}^{3 \times 3}$ defined in world space, opacity $o_i \in \mathbb{R}^1$

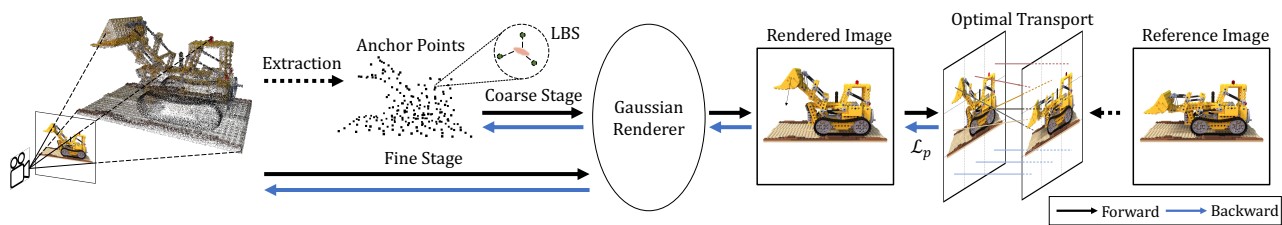

**Figure 2: An overview of our method. We address the single-image-driven editing task by an iterative gradient descent process that optimizes the 3D Gaussians to align with the reference image. To model long-range object deformation, we introduce the positional loss. To preserve the geometric consistency of the objects, we propose an anchor-based as-rigid-as-possible regularization scheme, a coarse-to-fine optimization strategy, and an adaptive masking strategy to identify the non-rigid deformation parts.**

and RGB color $c_i \in \mathbb{R}^3$ as spherical harmonics (SH). An anisotropic Gaussian filter $G_i(x)$ can be written as

$$G_i(x) = e^{-\frac{1}{2}(x-\mu_i)^T \Sigma_i^{-1}(x-\mu_i)} \tag{1}$$

To ensure that $\Sigma_i$ is always a positive semi-definite matrix during optimization, 3DGS formulates the covariance matrix as $\Sigma_i = R_i S_i S_i^T R_i^T$, with a 3D rotation matrix $R_i \in \mathbb{R}^{3\times3}$ represented by a quaternion $q_i \in \mathbb{R}^4$ and a scaling matrix $S_i$ represented by a 3D vector $s_i \in \mathbb{R}^3$.

When rendering an image of a specific view, 3DGS employs the EWA splatting method [58] to splat 3D Gaussians $G_i(x)$ to 2D Gaussians $G_i'(x) = \exp\left(-\frac{1}{2}(x-\mu_i')^T \Sigma_i'^{-1}(x-\mu_i')\right)$ onto the image plane. $\mu_i'$ is the center projection on the image plane and the 2D covariance matrix $\Sigma_i'$ of the splatted 2D Gaussian is given by

$$\Sigma_i' = JW\Sigma_i W^T J^T \tag{2}$$

Here, $J \in \mathbb{R}^{2\times3}$ is the Jacobian of the affine approximation of the perspective transformation. $W \in \mathbb{R}^{3\times3}$ represents the viewing transformation. Subsequently, 3DGS employs the alpha-blending method to aggregate the colors of Gaussians that cover the same pixel $u$

$$c = \sum_{i=1}^{N_u} \left(\prod_{j=1}^{i-1}(1-\alpha_j)\right)\alpha_i c_i \tag{3}$$

where $N_u$ is the number of overlapping Gaussians, and the alpha value $\alpha_i$ is formulated as $\alpha_i = o_i \cdot G_i'(u)$.

## 4 METHOD

Given the 3D Gaussian-based representation of a static scene and an edited image from a given viewpoint as the reference, the objective is to obtain the optimal 3D Gaussian parameters to align with the reference image. The involved editing operations may include translation, rotation, non-rigid geometric deformation, and texture change. A trivial approach is to use the gradient descent method to optimize the scene parameters, where the derivative of the pixel colors with respect to the 3D Gaussian parameters is given by the pixel-wise $L_1$ loss and the structure similarity (SSIM) loss as in the original 3DGS method [22]. However, these losses only generate intrinsically local derivatives, thus becoming less effective for optimizing long-range object translation and deformation and constraining the editing capability.

We draw inspiration from the success of DROT [50] in inverse rendering and introduce positional derivatives into the 3D Gaussian editing problem to capture long-range object motion. Leveraging the results of optimal transport (OT), we design a positional loss to explicitly capture long-range motions and guide 3D Gaussians movements. We back-propagate the positional derivatives to scene parameters via reparameterization, as detailed in Section 4.1. Some 3D Gaussians may be occluded when rendering the scene from the given viewpoint. To regularize the geometry of those occluded parts, we propose an anchor-based as-rigid-as-possible (ARAP) regularization method and adopt a coarse-to-fine optimization strategy for better convergence in Sec. 4.2. Furthermore, we design a novel adaptive masking scheme to identify and model non-rigid deformation parts in Sec. 4.3, thereby enabling better modeling of fine-grained details. We summarize the loss functions in Sec. 4.4. Fig. 2 illustrates the overview of our method.

### 4.1 Positional Derivative

To address potential long-range object translation and deformation, our key idea involves capturing the inherent 3D deformation field of the scene during editing. Therefore, we can explicitly guide the deformation and translation of 3D Gaussians during the optimization process. However, the 3D dense correspondence between the initial scene and those of the edited scene is unknown, and thus we cannot directly acquire the motion vector of a 3D point $p$. Inspired by DROT [50], we project the 3D field onto the image plane and leverage optimal transport to estimate 2D motion vectors.

Specifically, let $u \in \mathbb{R}^2$ denotes the 2D position on the image plane, and $c \in \mathbb{R}^3$ is its color. The vanilla 3DGS optimizes the learnable parameters $\theta$ of 3D Gaussians with the photometric loss $\mathcal{L}_c$, written as

$$\frac{\partial \mathcal{L}}{\partial \theta} = \frac{\partial \mathcal{L}_c}{\partial c}\frac{\partial c}{\partial \theta} \tag{4}$$

We extend the photometric loss $\mathcal{L}_c$ with a positional loss $\mathcal{L}_u$ defined on the 2D position $u$ to capture the motion of its corresponding local geometry in the inherent 3D space, and reformulate Eq. 4 by

$$\frac{\partial \mathcal{L}}{\partial \theta} = \frac{\partial \mathcal{L}_c}{\partial c}\frac{\partial c}{\partial \theta} + \frac{\partial \mathcal{L}_u}{\partial u}\frac{\partial u}{\partial \theta} \tag{5}$$

Here, $\mathcal{L}_u$ is defined as the difference between the 2D position $u$ in the original state and its corresponding position in the target state. Intuitively, $-\partial\mathcal{L}_u/\partial u$ indicates the movement direction of the local geometry around the 2D projected position $u$ with the goal

of reaching the state that matches the target image, while $\partial u / \partial \theta$, which can be further decomposed into $\partial u / \partial p \cdot \partial p / \partial \theta$, enables the differentiable optimization of scene parameters.

We treat the pixel centers as samples $u$ of the 3D field projected to the 2D image plane and leverage optimal transport to estimate the 2D correspondence. Then we define the transportation cost $w_{u,v}$ from pixel $u$ of the rendered image to pixel $v$ of the target image as a weighted sum of their color distance and positional distance.

$$w(u, v) = \lambda ||c(u) - c(v)||_2^2 + (1 - \lambda)||u - v||_2^2 \qquad (6)$$

where $\lambda$ is used to balance the two terms. After obtaining the dense 2D correspondences by optimal transport, the positional loss $\mathcal{L}_u$ is reformulated as the positional distance between pixel $u$ and its corresponding target $v$. At this point, the derivatives $\partial \mathcal{L}_u / \partial u$ can be directly deduced from the definition of $\mathcal{L}_u$, leaving $\partial u / \partial p$ and $\partial p / \partial \theta$ for us to calculate.

For the first term, according to Eq. 3, the color of pixel $u$ is computed by aggregating the colors of multiple Gaussians that cover the pixel, where the weight coefficient $\alpha_i \prod_{j=1}^{i-1}(1 - \alpha_j)$ measures the contribution of each 2D Gaussian $G_i'$ on the pixel $u$. To reduce computational costs, we reuse the intersection point $p_{u,i}$ of a 2D Gaussian $G_i'$ and a pixel $u$ as a sampling point when modeling the motion field of local geometry. We subsequently calculate the effect of positional derivatives $\partial \mathcal{L}_u / \partial u$ on the sampling point $p_{u,i}$ by

$$\frac{\partial u}{\partial p_{u,i}} = \alpha_i \prod_{j=1}^{i-1}(1 - \alpha_j) \qquad (7)$$

In the second term, note that the sampling operation that associates the intersection point $p_{u,i}$ and the properties of 2D Gaussian $G_i'$ is not differentiable, breaking the back-propagation of gradients. To back-propagate the gradients, we adopt the reparameterization method when drawing samples from the Gaussian distributions. Considering $p_{u,i}$ denotes a sample from a 2D Gaussian $G_i'$ with its center $\mu_i'$ and covariance matrix $\Sigma_i'$, we can view the sampling operation as a deterministic transformation of parameters $\mu_i', \Sigma_i'$ and a random variable $\epsilon \sim \mathcal{N}(0, I)$

$$p_{u,i} = \mu_i' + {\Sigma_i'}^{\frac{1}{2}} \epsilon \qquad (8)$$

Hence, the positional derivatives with respect to the center $\mu_i'$ and covariance matrix $\Sigma_i'$ of 2D Gaussian $G_i'$ can be given by

$$\frac{\partial p_{u,i}}{\partial \mu_i'} = I, \quad \frac{\partial p_{u,i}}{\partial \Sigma_i'} = \frac{\partial p_{u,i}}{\partial {\Sigma_i'}^{\frac{1}{2}}} \frac{\partial {\Sigma_i'}^{\frac{1}{2}}}{\partial \Sigma_i'} \qquad (9)$$

where $\partial p_{u,i} / \partial {\Sigma_i'}^{\frac{1}{2}}$ can be calculated using the reparameterization in Eq.8. ${\partial \Sigma_i'}^{\frac{1}{2}} / \partial \Sigma_i'$ can be obtained in closed form.

Inspired by 3DGS, which uses a tile-based rasterizer to achieve fast rendering, we propose a tile-based optimal transport matching to achieve high efficiency. Specifically, we split the screen into $16 \times 16$ tiles, average the colors of pixels within the same tile, and use Sinkhorn [11] divergence to approximate the positional derivatives between the downsampled images. Then, we can update the parameters of Gaussians using Eq. 7 and Eq. 9.

To demonstrate the influence of positional loss on long-range object deformation, we visualize the derivatives with respect to the

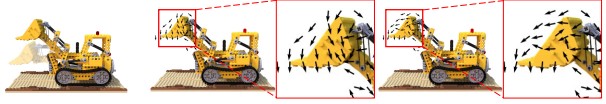

Put down the shovel    Ours    Gradient directions    L1&SSIM    Gradient directions

**Figure 3: Visualization of the gradients with respect to the centers of each Gaussian. The position loss can provide consistent and dense gradients to move down the bulldozer's shove, while the photometric losses only produce intrinsically local derivatives for optimization.**

centers of Gaussians and show the results in Fig. 3. Compared with the photometric losses adopted in 3DGS, our method can accurately determine the gradient descent direction to drive the blade of the Bulldozer downward.

## 4.2 Anchor-Based Deformation

In Eq. 7, the positional derivatives vanish as the weight coefficients go zero, thus failing to regularize occluded Gaussians at the reference view. As a result, only the visible parts of the involved objects are affected by the edited image, leading to structural discontinuity and breakdown. Motivated by the observation that the involved editing operations for real-world tasks are often sparse, spatially continuous, and locally rigid, we regularize the motions of 3D Gaussians with a local as-rigid-as-possible (ARAP) assumption as follows.

$$\mathcal{L}_{\text{arap}} = \frac{1}{N} \sum_i^N \sum_{j \in \mathcal{K}_i} \kappa_{ij} ||\overline{R}_i(\mu_i - \mu_j) - (\overline{\mu}_i - \overline{\mu}_j)||_2^2 \qquad (10)$$

Here, $\mu_i$ denotes the initial position of Gaussian $G_i$. $\overline{\mu}_i$ and $\overline{R}_i$ present the position and rotation at the current iteration, respectively. $\mathcal{K}_i$ represents the K-nearest neighbors (KNN) of $G_i$ and regularization weight $\kappa_{ij}$ is defined by the relative distance $d_{ij}$ between two Gaussians, $G_i$ and $G_j$, using Radial Basis Function (RBF), formulated as

$$\kappa_{ij} = \frac{\hat{\kappa}_{ij}}{\sum_{j \in \mathcal{N}_i} \hat{\kappa}_{ij}}, \text{ where } \hat{\kappa}_{ij} = \exp(-\gamma d_{ij}^2) \qquad (11)$$

where $\gamma$ is a hyper-parameter.

However, the ARAP term is defined within a small local region, generating non-zero gradients only when neighboring Gaussians undergo rotation or translation. Consequently, a substantial number of iterations is required to propagate regularization gradients to all occluded parts according to the movements of the neighboring visible parts. This can result in undesired deformation and suboptimal convergence during optimization. To address this issue, we propose to derive sparse anchor points from 3D Gaussians and then leverage them to capture the underlying 3D deformation field, substantially reducing the number of iterations compared to directly using 3D Gaussians.

Specifically, we voxelize the 3D scene and then compute the mass centers of 3D Gaussians in each voxel to extract a dense point cloud that covers the scene. We apply farthest point sampling (FPS) on the dense point cloud to downsample $N_a$ points and treat them as the initial anchor points $\{a_j\}_{j=1}^{N_a}$, where $a_j \in \mathbb{R}^3$ denotes the learnable positions of anchor point $j$ and $N_a$ is the number of anchor points. Each anchor point $a_j$ is also associated with a learnable

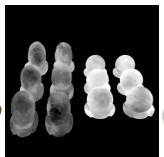 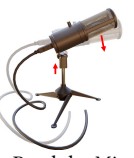 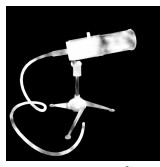

Stretch the balls   Distance Mask   Bend the Mic   ARAP Mask

**Figure 4: Adaptive rigidity masks. "Distance Mask" and "ARAP Mask" denote the learnable masks of the relative distance regularization term and ARAP regularization term, respectively.**

rotation matrix $R_j^a \in \mathbb{R}^{3\times3}$ represented by a quaternion $r_j^a \in \mathbb{R}^4$, which can be locally interpolated to yield a dense deformation field of the Gaussians. Instead of directly optimizing the position and rotation of Gaussians in each iteration, we optimize the parameters of anchor points to model the deformation field. After obtaining the anchor points, we can derive the deformation field of the Gaussians using linear blend skinning (LBS) [44] by locally interpolating the transformations of their neighboring anchor points. More details can be found in our supplementary materials.

Leveraging a set of sparse anchor points to model the complex deformation space may not faithfully align the scene with the target image. Therefore, we propose a coarse-to-fine optimization strategy to enhance visual quality. In the coarse stage, we utilize an anchor-based structure to optimize the position and rotation of anchor points, effectively capturing long-range changes. Subsequently, in the fine stage, we discard the anchor points and directly optimize both geometric and color parameters of each Gaussian. This approach helps mitigate artifacts such as noise on object boundaries and enhances the modeling of fine texture details. We employ the as-rigid-as-possible loss function on the anchor points during the coarse stage and on the 3D Gaussians during the fine stage.

### 4.3 Adaptive Rigidity Masking

In Eq. 10, ARAP assumes equal rigidity among the neighboring Gaussians of each Gaussian. However, in the real world, different parts of the 3D scene typically exhibit varying degrees of rigidity. Consider a T-pose human model: if we treat the rigidity of its joints and bones equally, undesired bending of bones may occur during deformation. Based on this observation, we incorporate an adaptive rigidity masking mechanism to help identify the extent of non-rigid deformation and mitigate the effects of rigid regularization.

Formally, we introduce a learnable mask $m_{ij} \in \mathbb{R}$ to each regularization weight $\kappa_{ij} \in \mathbb{R}$ and rewrite Eq. 11 as

$$\kappa_{ij}^m = \frac{\hat{\kappa}_{ij}}{\sum_{j\in\mathcal{N}_i}\hat{\kappa}_{ij}} \cdot \sigma(m_{ij}) \tag{12}$$

where $\sigma$ is the sigmoid function. Notably, the ARAP loss combines both relative rotation and relative distance regularization between Gaussians or anchor points. However, real-world object changes sometimes involve only one of these aspects. For instance, when we lower the blade of a Lego bulldozer, there is a relative rotation between Gaussians near the joint, while their relative geodesic distance remains unchanged. Therefore, we propose a rotation loss and a distance loss to provide explicit supervision on the rotations and

positions of Gaussians, respectively. We employ adaptive weights on the regularization terms in non-rigid regions, formulated as:

$$\mathcal{L}_{\text{rot}} = \frac{1}{N}\sum_i^N \sum_{j\in\mathcal{K}_i} \kappa_{ij}^{m^r} ||\bar{q}_i - \bar{q}_j||_2^2 \tag{13}$$

$$\mathcal{L}_{\text{dist}} = \frac{1}{N}\sum_i^N \sum_{j\in\mathcal{K}_i} \kappa_{ij}^{m^d} \left| |\bar{\mu}_i - \bar{\mu}_j|_2^2 - |\mu_i - \mu_j|_2^2 \right| \tag{14}$$

Here, $m_{ij}^d \in \mathbb{R}$ and $m_{ij}^r \in \mathbb{R}$ denote the learnable weight mask applied on the Gaussians for rotation and distance regularization, respectively.

Notably, the optimization process may fall into a trivial solution when the rigidity mask $m_{ij}, m_{ij}^d, m_{ij}^r$ approaches negative infinity. Thus, we periodically reset the weight masks $m_{ij}, m_{ij}^d, m_{ij}^r$ by taking the maximum value between the weight and a hyper-parameter $\eta$.

$$m_{ij} = \sigma^{-1}(\max(\sigma(m_{ij}), \eta)) \tag{15}$$

We visualize the learnable rigidity masks in Fig. 4, the masks of distance regularization term for the stretched material balls, and the masks of ARAP for the joint of microphone adaptively approach zero after optimization, illustrating the non-rigid deformation part in the scene.

### 4.4 Loss Function

In addition to the positional loss $\mathcal{L}_p$ described in Sec. 4.1, we also employ the photometric losses in 3DGS [22] to define the matching loss $\mathcal{L}_{\text{match}}$. We use $\mathcal{L}_{\text{match}}$ to generate gradients from the differences between the rendered image and the target image, written as

$$\mathcal{L}_{\text{match}} = \mathcal{L}_{\text{p}}(\mathbf{I}, \mathbf{I}^{\text{ref}}) + \lambda||\mathbf{I} - \mathbf{I}^{\text{ref}}||_1 + \lambda_{\text{SSIM}}\mathcal{L}_{\text{SSIM}}(\mathbf{I}, \mathbf{I}^{\text{ref}}) \tag{16}$$

For the learnable masks that adaptively identify the extent of the non-rigid deformation of each part, we apply an L1 regularization term to prevent degradation to zero.

$$\mathcal{L}_{\text{mask}} = \sum_i \sum_{j\in\mathcal{N}_i} |\sigma(m_{ij}) - 1| \tag{17}$$

The final loss of the coarse stage can be written as

$$\mathcal{L} = \mathcal{L}_{\text{match}} + \lambda_{\text{arap}}\mathcal{L}_{\text{arap}} \tag{18}$$
$$+ \lambda_{\text{rot}}\mathcal{L}_{\text{rot}} + \lambda_{\text{dist}}\mathcal{L}_{\text{dist}} + \lambda_{\text{mask}}\mathcal{L}_{\text{mask}} \tag{19}$$

For the fine stage, we additionally regularize the scales of each Gaussian in geometric editing and the colors of each Gaussian in texture editing, written by

$$\mathcal{L}_{\text{scale}} = \sum_i \left|\frac{\exp(\bar{s}_i)}{\exp(s_i)} - 1\right|, \quad \mathcal{L}_{\text{color}} = \sum_i \left|\frac{\sigma(\bar{c}_i)}{\sigma(c_i)} - 1\right| \tag{20}$$

## 5 EXPERIMENT

Due to the lack of publicly available benchmarks, we conducted quantitative experiments on the NeRF Synthetic (NS) Dataset [34] and the 3D Biped Cartoon Dataset [32], both of which contain the ground truth meshes of the reconstructed scenes. Specifically, we chose a viewpoint as the reference view to render an image for each scene in the NS dataset and the MipNeRF360 dataset. We

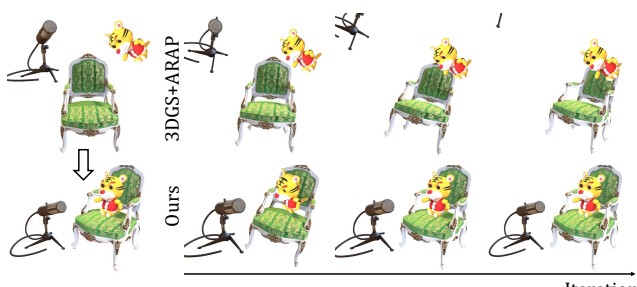

**Figure 5: Illustration of the optimization process for long-range rigid transformation. Compared with 3DGS supervised by the photometric losses, our method can handle long-range object movement well.**

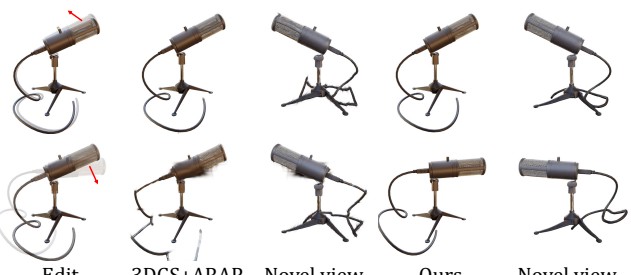

| Edit | 3DGS+ARAP | Novel view | Ours | Novel view |

**Figure 6: Geometric editing under different scales. Our method achieves a consistently better editing performance, especially on large-scale deformation.**

edited them using Adobe Photoshop to construct a reproducible benchmark for reference view alignment evaluation. The 3DBiCar dataset contains 1,500 3D Biped Cartoon Characters, each of which has a T-pose mesh and a posed mesh. We selected 52 characters for evaluation and generated 50 random views of the T-pose mesh for training 3DGS. For testing, we rendered eight surrounding images of the posed mesh, reserving one image for editing, while the others were utilized for novel view synthetic evaluation. We used Peak Signal-to-Noise Ratio (PSNR), Structural Similarity Index Measure (SSIM), and Learned Perceptual Image Patch Similarity (LPIPS) as the metrics. To demonstrate the effectiveness of our method on real-world data, we also evaluated it on 5 scenes from the Mip-NeRF 360 Dataset [4] and the Tanks & Temples Dataset [23] for qualitative experiments. We performed single-view video tracking on 2 scenes from the Panoptic Studio Dataset [21], given that our method can drive the inherent 3D world to temporally consistently align with the frame image once the initial 3D Gaussians model is provided.

## 5.1 Long-range Deformation

We conducted two toy experiments to demonstrate the effectiveness and necessity of positional derivatives in handling long-range editing operations. We initialized the first scene containing 3 objects and adjusted the content to align with the reference image. We used the original 3DGS with the ARAP term as the baseline, where the ARAP term maintains the structural stability. The optimization process is shown in Fig.5. Leveraging the positional loss, our method can drive objects to their target positions even if there is no overlap between their initial states and target states, such as

| Method | NeRF Synthetic | | | 3DBiCar | | |
| --- | --- | --- | --- | --- | --- | --- |
| | PSNR↑ | SSIM↑ | LPIPS↓ | PSNR↑ | SSIM↑ | LPIPS↓ |
| 3DGS+ARAP | 26.20 | 0.943 | 0.084 | 21.09 | 0.936 | 0.083 |
| DROT+ARAP | 20.70 | 0.834 | 0.169 | 15.59 | 0.901 | 0.135 |
| Ours | **35.00** | **0.970** | **0.042** | **24.62** | **0.955** | **0.053** |

**Table 1: Comparisons with other methods on geometric editing. We show the average PSNR/SSIM/LPIPS for reference view alignment on the NS dataset and novel view synthesis on the 3DBiCar dataset. ARAP denotes the as-rigid-as-possible regularization.**

the microphone and the toy tiger. In contrast, the baseline moves the microphone outside the screen, leading to sub-optimal convergence. We also tested the robustness of our method to non-rigid deformation under different scales. As shown in Fig.6, for short-range deformation, both 3DGS and our method can recover the deformation correctly. However, only our method can capture large deformations well.

## 5.2 Geometry Editing

We compared our method with DROT [50], which optimizes the position of mesh vertices obtained from NeRF2Mesh [45], and Deforming-NeRF [51], which models deformation by manually adjusting the deformable cage extracted from NeRF. As shown in Fig. 7, our method achieves precise alignment with the reference image, maintaining 3D consistency through the anchor-based structure and the two-stage optimization strategy. However, for DROT, the occluded parts require more iterations to back-propagate gradients from visible parts, leading to structural instability and undesired deformation, such as in the back of the drums. Deforming-NeRF faces limitations due to the resolution of deformable cages, particularly struggling with tasks like stretching objects such as hot dogs.

We also demonstrate the results of scene-level editing in Fig. 8. For scene-level editing, we first select a region of interest and render the image from a specific perspective. Then we can apply various 2D edits and back-propagate to the underlying 3D to align with these edits.

Since Deforming-NeRF requires manual adjustment of the cage, which is impractical to test on a large dataset, we quantitatively compare our method with vanilla 3DGS and DROT, and provide the results of reference view alignment and novel view synthesis in Tab. 1. Our method outperforms other methods in both tasks, exhibiting a consistent and significant improvement in metrics.

## 5.3 Hybrid Editing

Fig. 9 illustrates hybrid editing cases where we move the black pillar of the LEGO forward, elongate the cockpit, draw an MM logo on the side, stretch the chair horizontally, and draw an ACM logo on the back of it. We optimize the position and rotation of the anchors in the coarse stage to model long-range deformation, while in the fine stage, we refine the parameters of each Gaussian, including both geometry and color parameters. It can be observed that even for complex editing scenarios, our method consistently delivers promising results, demonstrating its robustness.

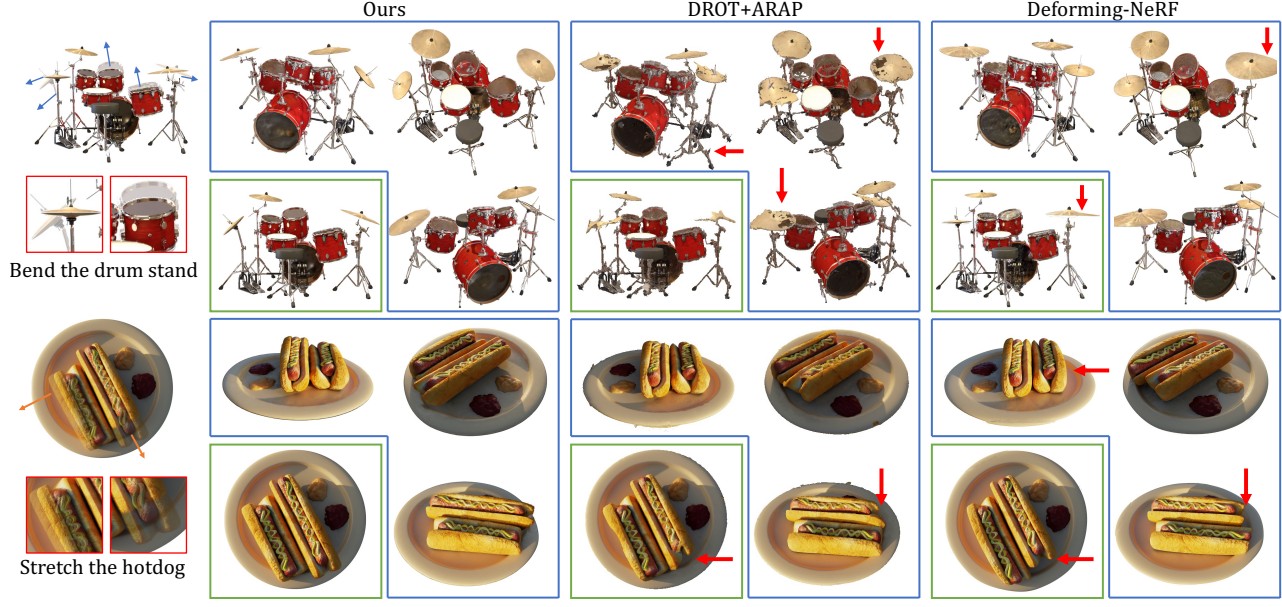

**Figure 7: Geometric editing on NS dataset. Green indicates the reference view of the edited image, and blue indicates novel views. Our method better aligns with the reference image while maintaining 3D consistency and structural stability.**

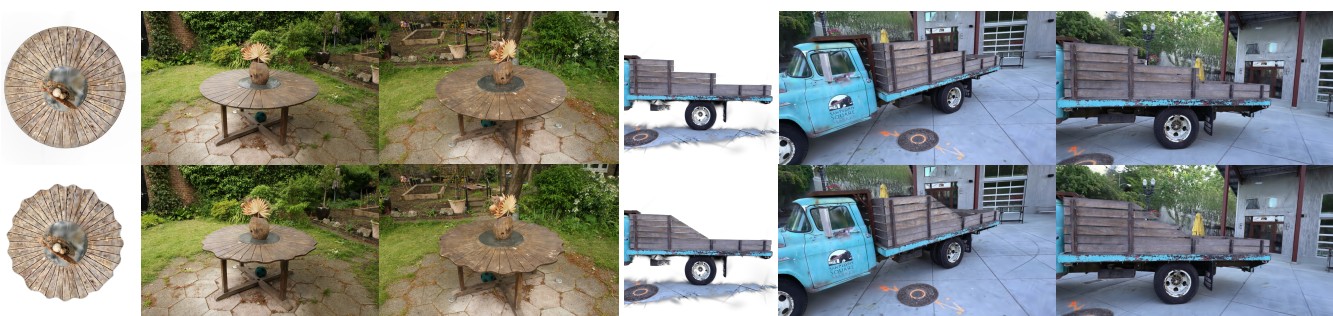

**Figure 8: Geometric editing on Mip-NeRF 360 Dataset. We wavy the edges of the table in the garden and slope the planks of the truck. Our method aligns well with the reference image while maintaining 3D consistency and structural stability.**

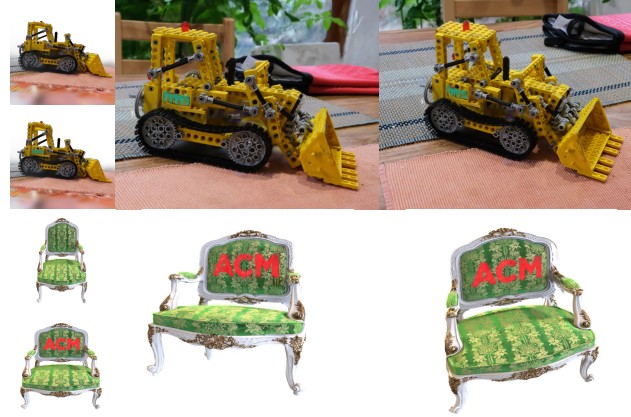

**Figure 9: Hybrid geometry and texture editing. Our method enables simultaneous editing of geometry and textures in a single optimization process.**

## 5.4 Single-View Video Tracking

Given the initial 3D Gaussian scene reconstructed from multi-view images, our method enables us to use a single-view video to track the underlying dynamic 3D scene by aligning the rendered image with the subsequent video frames. We only use the coarse stage and optimize the position and rotation of the anchors for fast convergence. We show the reference video frame and two novel views in Fig. 10. Our method can capture the long-range object motion and maintain both spatial and temporal consistency, producing promising novel view synthesis results.

## 5.5 Ablation Study

We conducted ablation studies on positional loss, two-stage optimization, adaptive rigidity masking, and explicit supervision of relative rotation (Eq. 13) and distance (Eq. 14). The results are summarized in Table 2, providing quantitative insights into the effectiveness of each component. Apart from the explicit regularization

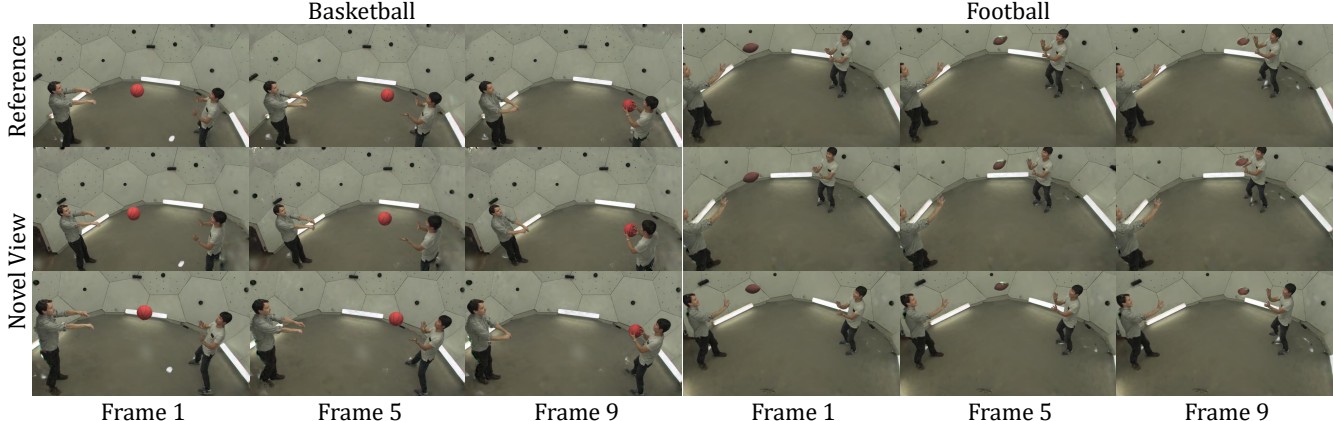

Basketball                                          Football

Reference

Novel View

Frame 1          Frame 5          Frame 9          Frame 1          Frame 5          Frame 9

**Figure 10: Single view video tracking. Given the initial 3D scenes reconstructed from multi-view images, our method can capture the dynamic 3D scene using single-view video and produce consistent novel view synthesis results.**

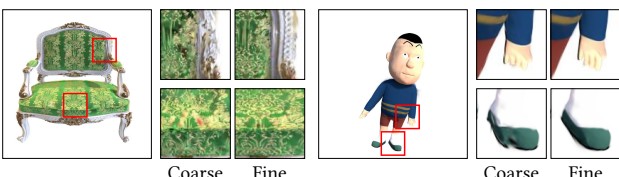

Coarse    Fine                          Coarse    Fine

**Figure 11: Comparison of the optimized results after coarse stage and fine stage. The coarse stage mainly captures long-range translation and deformation, while the fine stage achieves fine-grained texture and geometry reconstruction.**

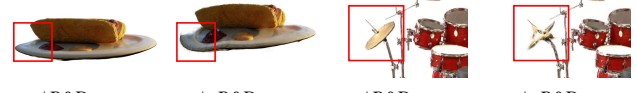

w/ R&D terms    w/o R&D terms    w/ R&D terms    w/o R&D terms

**Figure 12: Ablation study of the relative rotation and distance (R&D) regularization terms.**

| Method | NeRF Synthetic | | | 3DBiCar | | |
|---|---|---|---|---|---|---|
| | PSNR↑ | SSIM↑ | LPIPS↓ | PSNR↑ | SSIM↑ | LPIPS↓ |
| 3DGS+ARAP | 26.20 | 0.943 | 0.084 | 21.09 | 0.936 | 0.083 |
| + Position | 30.99 | 0.953 | 0.064 | 23.42 | 0.946 | 0.065 |
| + Anchor | 34.92 | 0.974 | 0.035 | 24.01 | 0.952 | 0.057 |
| + Mask | **36.20** | **0.977** | **0.032** | 24.33 | 0.951 | 0.058 |
| + R&D(Full) | 35.00 | 0.970 | 0.042 | **24.62** | **0.955** | **0.053** |

**Table 2: Ablation studies of different components. "Position" denotes the position loss computed by Optimal Transport. "Anchor" denotes the anchor-based deformation and two-stage optimization. "Mask" and "R&D" are the learnable rigidity mask of ARAP loss and explicit regularization of relative rotations and distances, respectively.**

of relative rotations and distances, the addition of any other components consistently leads to noticeable improvements in target view alignment and novel view synthesis. Moreover, explicit regularization helps maintain structural stability, prevents overfitting to the reference view, and enhances the rendering quality from other perspectives.

Fig. 11 presents the optimization results of the coarse stage and fine stage to provide a better understanding of anchor-based deformation and coarse-to-fine optimization. The coarse stage captures long-range deformation during editing and aligns the 3D scene roughly with the reference image, while the fine stage reduces artifacts on the object boundaries and models fine texture details, thereby achieving better alignment.

Additionally, we offer a visual comparison of ablating the explicit regularization term of the positions and rotations in Fig. 12. Notably, explicitly regularizing the relative rotation and position between two neighboring Gaussians can effectively address needle-like problems and reduce structural errors from a new perspective.

## 6 CONCLUSION AND LIMITATION

We present a single-image-driven 3D scene editing approach that enables intuitive and detailed manipulation of 3D scenes. We address the problem through an iterative optimization process based on 3D Gaussian Splatting. To handle long-range object translation and deformation, we introduce positional loss into 3D Gaussian scene editing and differentiate the process through reparameterization on 2D Gaussians. To maintain the geometric consistency of the occluded Gaussians in the edited image, we propose an anchor-based As-Rigid-As-Possible (ARAP) regularization and a coarse-to-fine optimization strategy. Additionally, we design a novel rigidity masking strategy to achieve precise modeling of fine-grained details. Experiments demonstrate our superior editing flexibility and quality compared to previous approaches.

Our method has the following limitations. Since our method leverages optimal transport to calculate the positional loss, it is limited by the accuracy of pixel matching. In areas with weak texture information, where most of the rendered pixels are similar, the Sinkhorn divergence[14] may fail to provide a correct match, thus affecting the optimization of the underlying 3D scene. Additionally, since our method prefers driving 3D Gaussians rather than growing and pruning, it limits the resolution in texture editing. Disentangling geometry and texture, as proposed in [52], may improve the quality of texture editing.

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
