# OpenReview forum: "3D Gaussian Editing with A Single Image"
_acmmm.org/ACMMM/2024/Conference — MM2024 Poster_

### Official Review · Reviewer_iy6F · 2024-05-20

**Rating:** 5
**Confidence:** 3

**Summary:**

The paper proposes a 3DGS editing method given a single user-specified image, with experiments of geometry editing (especially for long-range deformation), and texture editing.

**Strengths:**

The paper introduces motion vectors with position loss in the image domain, providing a novel approach to GS editing. An ARAP loss regularizes GS to maintain structural consistency, while an adaptive rigidity mask identifies the extent of non-rigid deformation. These losses are designed in an explainable and intuitive manner, resulting in a complete pipeline.

**Limitations:**

1. **Lacking Related Works in NeRF Editing**: The paper should include more references to related works in NeRF editing. A series of studies [1-6] incorporate UV mapping to enable local and pixel-level editing in NeRF. Including these references would provide a more comprehensive background and context for the current work.

2. **Issues with Single Image-Based Editing**: The single image-based editing method struggles with consistent novel image synthesis, as seen in the supplementary video (geometry editing/3dbicar/39, 2075). This observation undermines the claim that other methods heavily rely on accurate mesh manipulation, which guarantees novel view synthesis. This issue raises concerns about the robustness of the proposed method when editing objects with more complex geometry. However, I still believe this approach is novel and offers significant guidance for the GS editing domain, as using a single image specified by users is an easy and intuitive way to perform editing.

3. **Comparison with Other Methods**: While some GS-editing works are still under review, it would be better to compare the proposed method with them (e.g., GaussianEditor, CVPR'24). If not, comparisons with NeRF-based editing methods such as NeuMesh and Seal-3D, which also use a single image as input, would help evaluate the performance of the proposed method. These comparisons would provide a better understanding of the proposed method's strengths and weaknesses relative to existing approaches.

4. **Time efficiency is also a concern.**.  While GS achieves real-time rendering, it is important to assess whether your editing method allows for interactive manipulation and real-time rendering of novel images after editing.

5. Can the authors provide visual examples for their ablation studies on the loss terms? While the numbers in Table 2 are metrics of supervised edited image, the metric alone is not convincing. Comparative images from novel and different views would better illustrate the contribution of each loss. Additionally, showing some failure cases due to textureless areas or mismatched pixels would be beneficial. This would help in understanding the limitations and robustness of the proposed method.

Reference:
[1] NeuTex: Neural Texture Mapping for Volumetric Neural Rendering

[2] Neural Parameterization for Dynamic Human Head Editing

[3] Parameterization-driven Neural Implicit Surfaces Editing

[4] Nuvo: Neural UV Mapping for Unruly 3D Representations

[5] Learning an Isometric Surface Parameterization for Texture Unwrapping

[6] Texture-GS: Disentangling the Geometry and Texture for 3D Gaussian Splatting Editing

[7] GaussianEditor: Swift and Controllable 3D Editing with Gaussian Splatting

**Suitability:**

3

---

### Official Review · Reviewer_k7cf · 2024-05-24

**Rating:** 6
**Confidence:** 3

**Summary:**

This paper proposes a novel 3D scene editing approach for 3D Gaussian Splatting using a target image as a reference. Specifically, it aligns 3D Gaussians with the target image through gradient-based optimization, introducing optimal transport and a positional loss to capture long-range motions. The paper also designs an anchor-based as-rigid-as-possible regularization method and a coarse-to-fine strategy to ensure geometric consistency. Furthermore, a masking mechanism is introduced to extend non-rigid deformation. Experimental results demonstrate that the proposed method achieves significant quality improvements when geometrically editing 3D scenes.

**Strengths:**

1.	The paper is well-written and easy to follow, with a good structure.
2.	The proposed method is technically sound and thoroughly demonstrated.
3.	The experiments are sufficient and fully demonstrate the performance of the proposed methodology.

**Limitations:**

1.	It would be better to showcase the time complexity, as the proposed method introduces several terms to restrict the optimization process.
2.	In section 4.2, it is not clear when to derive sparse anchor points from 3D Gaussians. Furthermore, the operation to “compute the mass centers of 3D Gaussians in each voxel” is not clearly explained.
3.	Considering editing operations like non-rigid geometric deformation, can the proposed method outperform editing methods using text-to-image models? This would be an interesting comparison to explore.

**Suitability:**

3

---

### Official Review · Reviewer_BkGy · 2024-05-24

**Rating:** 4
**Confidence:** 3

**Summary:**

This paper propose a single-image-driven 3D scene editing approach based on 3D Gaussian Splatting. 3D Gaussian can be deformed through mapping an image deformation back to the 3D scene.

**Strengths:**

It is an interesting and effective way to deform 3D Gaussian by manipulating a 2D viewpoint. It is able to manipulate the 3D model from different 2D views.

The paper is well-written. The experiments demonstrates various applications, such as geometry editing, long range deformation and hybrid editing.

**Limitations:**

The time performance is not discussed. Does the method run in real time? The time performance is critical for an editing task.

In the video, I can see artifacts during editing. The reasons are not clear.

I would like to see some geometric detail editing. There should be some stress tests to illustrate the upper bound for details deformation. This could strengthen the work.

**Suitability:**

2

---

### Official Review · Reviewer_uA8r · 2024-05-25

**Rating:** 3
**Confidence:** 3

**Summary:**

This paper proposes a method to edit a 3D Gaussian representation of a scene by editing an image rendered from a single viewpoint and constraining the Gaussuans to adapt to the edited image with acoarse to fine optimization scheme with ARAP constraints to capture long range deformations.
To adapt to the presence of non rigid parts a masking strategy is adopted

**Strengths:**

The method is able to transfer reasonably shapes and textures edited in 2D to the 3D Gaussians set allowing to obtain a coherent scene.The method is compared for local editng of NERFs.
The methods seem to correct artifacts that a straightforward optimization of the rendering loss with the edited image create.

**Limitations:**

The description of the adaptive rigidity mask is not clear.
Actually the whole editing pipeline seems limited. The first step in editing both from 2D single views and 3D editors would be to perform a segmentation of the object in the scene allowing to decouple the modified Gaussians from the unchanged ones. This is fundamental and not addressed. It should also be not too difficult, as it is possible to do it manually or automatically on the 3D Gaussian set (or starting on the image and propagating on the Gaussians).
Actually the editing actions demonstrated would be easier to perform directly on the 3D Gaussians rather than in 2D. Typically in 3D editors, users selects 3D parts to be transformed on a 2D interface and the selection is possibly propagated in 3D. So for a 3D Gaussian editor I would expect that the interactive editing could start with the propagation of the selection of objects to be edited, pissibly smartly segmenting the object corresponding to the 3D part using shape and color information and then introducing editing actions mapped from 2D and 3D. It's hard for me to tink that is better to edit the raster image rendered and the optimize the whole Gaussian set to adapt with rigidity constraints that it is hard to generalize without having a semantic segmentation of the object.
The fact that only limited modifications on mostly segmented parts seems to confirm this.

**Suitability:**

3

---

### Meta-Review · Area_Chair_PeAS · 2024-07-08

**Recommendation:** Accept (Poster)
**Confidence:** 5

**Metareview:**

This paper presents a new single-image-driven 3D scene editing approach based on 3D Gaussian Splatting, enabling intuitive manipulation via directly editing the content on a 2D image plane. Both the method and task look interesting and new to some extent. All reviewers have positive final ratings, are satisfied with the response, and recommend accepting the paper. I agree with their recommendation. Thanks for the authors' effort and rebuttal.